# Mobile Applications in Mood Disorders and Mental Health: Systematic Search in Apple App Store and Google Play Store and Review of the Literature

**DOI:** 10.3390/ijerph19042186

**Published:** 2022-02-15

**Authors:** Sophie Eis, Oriol Solà-Morales, Andrea Duarte-Díaz, Josep Vidal-Alaball, Lilisbeth Perestelo-Pérez, Noemí Robles, Carme Carrion

**Affiliations:** 1Fundació HiTT (Health Innovation Technology Transfer), 08015 Barcelona, Spain; seis@fhitt.org; 2Canary Islands Health Research Institute Foundation (FIISC), 38109 Tenerife, Spain; andrea.duartediaz@sescs.es; 3Health Promotion in Rural Areas Research Group, Gerència Territorial de la Catalunya Central, Institut Català de la Salut, 08272 Barcelona, Spain; jvidal.cc.ics@gencat.cat; 4Unitat de Suport a la Recerca de la Catalunya Central, Fundació Institut Universitari per a la Recerca a l’Atenció Primària de Salut Jordi Gol i Gurina, 08007 Barcelona, Spain; 5Faculty of Medicine, University of Vic-Central University of Catalonia (UVIC-UCC), 08500 Vic, Spain; 6Evaluation Unit of the Canary Islands Health Service (SESCS), REDISSEC, 38109 Tenerife, Spain; lilisbeth.peresteloperez@sescs.es; 7eHealth Lab Research Group, School of Health Sciences and eHealth Centre, Universitat Oberta de Catalunya (UOC), 08035 Barcelona, Spain; nrobles@uoc.edu (N.R.); mcarrionr@uoc.edu (C.C.)

**Keywords:** mental health apps, mobile applications, mood disorders, bipolar, depression, dysthymia, Apple App Store, Google Play Store

## Abstract

Objectives: The main objective of this work was to explore and characterize the current landscape of mobile applications available to treat mood disorders such as depression, bipolar disorder, and dysthymia. Methods: We developed a tool that makes both the Apple App Store and the Google Play Store searchable using keywords and that facilitates the extraction of basic app information of the search results. All app results were filtered using various inclusion and exclusion criteria. We characterized all resultant applications according to their technical details. Furthermore, we searched for scientific publications on each app’s website and PubMed, to understand whether any of the apps were supported by any type of scientific evidence on their acceptability, validation, use, effectiveness, etc. Results: Thirty apps were identified that fit the inclusion and exclusion criteria. The literature search yielded 27 publications related to the apps. However, these did not exclusively concern mood disorders. 6 were randomized studies and the rest included a protocol, pilot-, feasibility, case-, or qualitative studies, among others. The majority of studies were conducted on relatively small scales and 9 of the 27 studies did not explicitly study the effects of mobile application use on mental wellbeing. Conclusion: While there exists a wealth of mobile applications aimed at the treatment of mental health disorders, including mood disorders, this study showed that only a handful of these are backed by robust scientific evidence. This result uncovers a need for further clinically oriented and systematic validation and testing of such apps.

## 1. Introduction

Currently, the majority of the world population uses a smartphone, and we use it an average of 61 h per week [1]. In fact, a 2018 report of the Spanish National Observatory for Telecommunications and the Information Society (ONTSI) reported that 78.9% of Spaniards (≥15 years of age) own a smartphone and 74.8% of them used them to access the Internet [2]. It is estimated that there are more than 325,000 applications (apps) classified as health or wellness applications. A more detailed analysis reveals that more than 50% of these apps received less than 500 downloads during the period 2016–2018 [3].

It seems there are no standard tools or procedures to empirically assess the safety and potential effectiveness or harm of all these apps. There exist limited or no governing bodies to oversee and regulate app development and availability, making access to trustworthy and validated apps unstructured and difficult. This is particularly problematic with mental health–related mobile apps, as many developers are not affiliated with mental health professionals. Moreover, many apps are not equipped to deal with potential mental health emergencies that may occur during app use, by, for example, displaying (correct) suicide hotline numbers [4,5]. Put together, all of this generates a lack of trust among the final users, that is healthcare professionals, patients, or health-conscious citizens concerned [6,7].

However, despite these drawbacks, health apps carry great potential. There are already several trials that evidence the positive impact of mobile health interventions on the management of various pathologies such as diabetes, asthma, or hypertension [8].

One of the most prevalent health problems in our society are mental health disorders. Currently, amid the COVID-19 pandemic, the number of people suffering from mental health disorders is significantly increasing. Recent studies show the effect of the pandemic on our lives. Physical health, economic crisis, confinement, social distancing, or burnout syndrome are beginning to have important impacts on the mental health of the global population, even on people without previous mental pathologies [9].

One of the main obstacles in the management of mental health pathologies is the stigma that people affected by these disorders must overcome. This influences a patient’s entire journey, starting with asking for help, until access to treatment, up to and including long-term follow-up, which may be necessary if disorders become chronic. Digital health tools can help both in the prevention of mental health disorders, as well as during their treatment, acting as co-adjuvant in the therapeutic approach while reducing stigma [10,11]. There are between 10,000 and 20,000 apps within the category of mental health available in the markets [12,13,14]; however, it is estimated that only about 3% to 4% are evidence-based [15]. Mobile health interventions can help promote self-care and empower patients [16] and thus avoid treatment limitations [17] and contribute to improvements in the symptoms and quality of life of people affected by depression and/or anxiety [18,19,20,21]. Since mobile health offers 24/7 support, patients may be more willing to communicate the presence of severe symptoms on technology platforms rather than face-to-face [22]. A handful of recent systematic reviews and meta-analyses have compiled existing evidence on the use of apps for the treatment of mental health. One overview of 9 systematic reviews found that mobile mental health applications (MHapps) showed some significant results in reductions of depression and stress scores [23]. A further meta-review of meta-analyses found that studies focusing on anxiety or depressive symptoms were of moderate to high quality and generally had small to medium effect sizes [15]. Lastly, a systematic review focusing on the treatment of bipolar disorder found that smartphone-based interventions and monitoring systems had a significant positive impact on both depressive and manic symptoms [24].

However, despite the “apptimism”, meaning the optimistic outlook on the potential benefits of app utilization, generated by apps in some sectors, mobile health is not yet a strategy that is commonly used in the management of health issues in general or mental health and mood disorders. Barriers to patient app use as well as to larger-scale adoption include concerns surrounding safety, credibility, unfamiliarity and ignorance, usability, personalization, as well as information governance [25,26,27,28]. Healthcare professionals also show some skepticism and lack of knowledge regarding what the best strategy for each patient may be. Some studies suggest that this could be due to difficulty in identifying the right applications [29] or the lack of evidence supporting its potential effectiveness [30]. In fact, often the applications best positioned in commercial repositories are not of the highest quality and neither have the most proven effectiveness [31].

Our study aimed to understand and characterize the current landscape of mobile applications available in the markets to manage mood disorders, in particular, such as depression, bipolar disorder, and dysthymia. The Diagnostic and Statistical Manual of Mental Disorders 5 (DMS-5) outlines criteria for the diagnosis of major depressive disorder, in which an individual must be experiencing five or more specific symptoms during at least 2 weeks, with at least one symptom being depressed mood or loss of interest or pleasure [32]. Dysthymia is a form of depression, also called persistent depressive disorder, and is described by the DSM-5 as a mood disturbance distinguished by low-grade depressive symptoms persisting for at least two years [32]. The National Institute of Mental Health (NIMH) describes bipolar disorder as “a mental disorder that causes unusual shifts in mood, energy, activity levels, concentration, and the ability to carry out day-to-day tasks” [33]. Three types of bipolar disorder are defined, and all involve clear changes in mood, energy, and activity levels. These include periods of extremely elevated, irritable, or energized behavior, referred to as manic episodes, and sad, depressed, indifferent, or hopeless periods, known as depressive episodes [32,33].

## 2. Materials and Methods

### 2.1. App Search and Filtering

To explore the landscape of MHapps for the treatment of mood disorders a list of key search terms was developed, namely: “bipolar”, “depression”, “dysthymia”, and “mood”. These were used to search both the Apple App Store and the Google Play Store. To extract the necessary data from the stores in a structured way, and then be able to filter resulting apps appropriately, an in-house search engine was developed. It produced spreadsheets containing the following information: App name; app developer; approximated number of downloads in Google and number of user ratings for Apple; average user rating; genre; language, in the Apple App Store only; price; release date; and the corresponding URL. Our tool was limited to extracting up to 200 of the most relevant apps for each search.

The application allows the search of key terms in the App/Play Store of a particular country, therefore gaining access to various, potentially region-specific apps. This research restricted its search to these countries: Spain, France, Germany, Italy, United Kingdom, and the United States. The search terms were used both in English and the country’s native language between the dates 30/11/2020 and 11/12/2020. In the cases of the UK and USA, English, and Spanish terms were used.

After generating spreadsheets of results for each search term, the results were compiled by country into master files and the initial two filters were applied: Only apps with >10,000 downloads; and only apps belonging to the genres “Health and Fitness”, “Lifestyle”, “Medical”, and “Education”. After removing any apps not complying with the above, all country-specific results were compiled into one overall master file. Repeated app results were manually removed, and two further filters applied: Excluding any apps explicitly not available in Spanish; and apps with a most recent update dating back further than a year, i.e., before 2020. This was done to try to exclude apps that were no longer actively used and/or improved upon. Only Apple provided specific language information, thus ones who did not list Spanish were removed. In the Google results some apps were only available in, for example, Portuguese, Italian, etc., and therefore were also removed.

Lastly, while systematically reviewing all apps, 4 distinct app-types arose, which were eliminated from the results due to irrelevance to the project. Namely: Meditation apps; journaling, dairies or (mood) tracker apps; diagnostic tests, with varying degrees of clinical legitimacy (e.g., filling in questionnaires of symptoms for categorization for depression, anxiety or bipolar, without intention of treatment); and apps providing information only, without any element of interaction or treatment. Although many of these apps may be used in the global treatment of mental disorders, their primary focus is not strictly on intervention, hence they were excluded. Table 1 provides an overview of the applied app exclusion criteria.

### 2.2. App Characterization

The filtered apps were then systematically analyzed in more detail to understand the following parameters: Specific type of app and objective; developer profile (who and what kind of background do they have); target population; type of data collected by the app.

### 2.3. Publication Search

Furthermore, a literature search was performed, to see whether any of the apps were supported by any type of scientific evidence on their acceptability, validation, use, effectiveness, etc. This was done by searching PubMed using each app’s name in combination with several keywords (displayed in Figure 1). No further filters or search criteria were applied to the PubMed search. For each relevant search result, any listed related literature was also examined to ensure the inclusion of all relevant available information that may not have been shown using the search terms in Figure 1. Additionally, each app’s developer’s website was examined for references to published evidence to also be included.

## 3. Results

The search resulted in >500 apps, of which 30 apps met the inclusion and exclusion criteria. Figure 2 shows the flowchart of app results obtained at various stages of filtering and application of inclusion and exclusion criteria.

Certain metrics were not readily available from the platforms, specifically, the number of app downloads from the Apple App Store. Therefore, as a proxy, the number of user ratings was used, with a more generous threshold of >100 ratings, as there tended to be far fewer ratings. The app results are split up into three categories of results, namely: 8 apps (26.7%) with published evidence available (Table 2); 15 apps (50%) with no published evidence, but legitimate background (Appendix A); and 7 apps (23.3%) with limited available information (Appendix A). More detailed information, including developer information, URLs, and the type of data collected, of the apps which had connected published evidence can be found in Appendix A.

The results include technical details (expanded on in Appendix A) of each app and information regarding the type of service it provides and under which objective, what its target population is, the type of data collected by the app, and the profile of the developer(s), i.e., their backgrounds and potential ties to research groups, clinicians, public health efforts, etc.

Appendix A details all the published articles that we were able to identify related to one of the apps listed in Table 2. A total of 27 publications were found through the PubMed search, supplemented by additional findings from developers’ websites. Each of these publications described distinct studies of varying sizes and robustness. The results contained: 7 observational/longitudinal studies; 6 randomized controlled trials (RCTs), including one crossover study; 5 nonrandomized feasibility/usability/acceptability/effectiveness studies, including 4 pilot studies; 4 descriptive studies; 1 user satisfaction survey; 1 case study; 1 focus group; 1 context analysis; and 1 study protocol. The studies showed a high degree of heterogeneity in terms of the type of investigated mental health disorder. Despite searching for apps designed to treat the mood disorders such as depression, dysthymia, and bipolar, these studies focused on various disorders that do not all fall into one of these categories yet could be understood to be interrelated in the larger context of mental health. Table 3 illustrates this heterogeneity and the various disease or treatment areas addressed in each publication, which mainly included depression and anxiety, but also obsessive-compulsive disorder (OCD), body image disorder (BID), among others. Some publications included patients of various disorders, such as both anxiety and depression patients. Lastly, other publications focused more on the usability, acceptability, or user satisfaction than any clinical benefit of the apps [34,35,36,51,52].

## 4. Discussion

### 4.1. Principal Findings

This article presents a systematic search of the Apple and Android app stores for mobile apps specifically targeted at mood disorders. In light of the fact that, despite limiting our search to mood disorder-related terms, the resulting apps and associated publications concerned disorders in the broader topic of general mental health, this discussion treats MHapps in general, not limited to mood disorders. This is because a lot of the discussed issues apply broadly to MHapps, as well as to other health apps.

While this study is not the first to examine the available health app-landscape, there exist only a limited amount of literature describing systematic searches of MHapps (e.g., [61,62,63,64]) and the empirical evidence base surrounding them. Other reviews of MHapps were focused on other country contexts (e.g., China [65], Arabic speakers [66]), as well as different, more general, or more specific disease areas or patient populations (e.g., chronic conditions in general [67,68], peripartum mood disorders [69], eating disorders [70], older adults [71], attention-deficit hyperactivity disorder [72]).

We aimed to explore not only the breadth and variety of available apps for mood disorders, but most importantly we sought to understand the scientific and clinical evidence base supporting the use of the most popular ones. Emphasis on evidence-based interventions could propel a pivotal paradigm shift away from more traditional ways of treating mental health disorders and toward mHealth/eHealth. Especially in a post-COVID era that has made us rethink conventional patient-practitioner interactions.

What we found, however, was that while the app marketplace offers users a wide range of apps marketed toward mental health intervention, only 30 mobile apps fit the inclusion criteria, and of those only 8 (26.7%) were supported by any type of published scientific evidence. Moreover, while the resulting apps could be used in the treatment of mood disorders, only 5 of the 27 associated publications measured the effectiveness of an MHapp on health outcomes and symptoms of mood disorders in particular, namely depression, including postpartum depression (PPD) [37,38,53,55,56]. Appendix A of the Appendix A provides an overview of all publications, the type of evidence reported within them, and their principal findings. Most applications in this overview are not supported by studies published in scientific journals and lack the approval of official agencies endorsing their use. The few publications which did report clinical outcome measures found positive changes to measures such as depression symptom severity, or improved well-being scores. However, studies were early stage and sample sizes tended to be small.

This implies that while a plethora of mobile interventions is developed and marketed, the channels through which this is commonly done do not lend themselves to implementation within healthcare provision contexts; contexts which usually rely on robust effectiveness and safety testing. This might mean that mHealth interventions developed and tested in formal research settings for research purposes are rarely made available to the general public, they do not garner popularity and attention, they are simply non-existent or that they are just widely outnumbered by applications developed in non-research settings.

Our findings highlight considerable shortcomings in the clinical validation of even the most popular MHapps. While the 5 aforementioned publications did report a promising amelioration of wellbeing or various depression symptoms, these findings arise from small-scale (pilot) studies, with various methodological limitations. Therefore, these findings cannot be considered robust enough to provide strong scientific support for the routine clinical use of such interventions. Moreover, comparing these trial findings with the type of robust and methodologically sound evidence necessary to approve, implement, and recommend other kinds of health-interventions (i.e., drugs, devices, therapeutic methods), once more underscore the lack of systematic testing and validation of mental health apps.

Although app developers have made efforts to incorporate evidence-based treatments, such as cognitive behavioral therapy, more research is needed to improve the clinical validity, treatment reliability, and safety of MHapps. This is supported by the findings of a recent systematic search and content analysis of depression apps, which assessed how mobile applications measured up against the National Institute for Health and Care Excellence (NICE) guidelines for the treatment of depression in adults [61]. None of the identified apps fully aligned with the NICE guidelines and authors urged developers to consult and regard relevant guidelines and standards throughout app development and content design.

This calls into question whether direct-to-consumer (DTC) is the most effective and safe route for MHapps to be marketed and distributed to patients struggling with complex mental health pathologies.

### 4.2. Safety and Ethical Considerations of MHapps

The principal area of concern regarding health apps in general, but of course also MHapps, is privacy and the use and protection of personal/medical data. A recent analysis of privacy-related permissions of diabetes apps found that approximately 60% of the analyzed apps requested potentially dangerous permissions, meaning permissions that might lead to data breaches and thus pose a considerable risk to data privacy [73]. Moreover, authors found that app users may not always realize that the business model of free apps is largely based on advertising and, consequently, on directly or indirectly sharing or selling their private data to unknown third parties [73]. These concerns about privacy further expand in the context of apps that use passive monitoring of individuals with mental illness. This involves collecting data from patients through sensors without requiring direct patient input, such as speech patterns, mobility, activity level and signs of social interaction [74,75]. A considerable population segment does not want their digital activity to be monitored and tracked, and without an understanding of the digital economy, which is based on creating value from the analysis of tracked behavioral data, encouraging the use of MHapps may inadvertently lead to harm [76,77,78,79,80,81].

Furthermore, a risk to MHapp users’ safety may be the promotion of unproven, unsafe, and misleading messages. A study of 61 frequently used MHapps concluded that the themes they emphasized may promote medicalization of normal mental states and imply individual responsibility for mental health [10]. While the idea of mental health care for everyone might help reduce stigma, this type of messaging could lead to overdiagnosis and pharmaceutical overtreatment [82] and be potentially dangerous for diagnosed patients who need a clear understanding of when to seek professional help [10]. Moreover, in the absence of adequate regulation and if affiliations to regulated mental health professionals are lacking, DTC MHapps may connect users to nonprofessional therapists or chatbots with limited personalized treatment capacities. MHapps may also fail to provide emergency information, all of which exacerbates concerns over safety, accountability, and treatment effectiveness and adherence [4,5,83,84,85].

### 4.3. Effectiveness and Evidence of MHapps

Previous studies corroborate our findings that despite the potential of mobile mental health intervention, only a slim percentage of MHapps are based on clinically validated research and a lack of evidence on the effectiveness of mobile health apps is pervasive [64,86,87,88]. Even reviews of MHapp controlled trials generally conclude that studies are of mixed quality and highlight the necessity for further systematic investigation [89]. However, considering the low cost of entry for app developers in general, it is unlikely that many of them will ever be able to afford even a simple clinical trial to validate effectiveness and safety [90]. This is especially true for private sector products, which will often not be subject to more rigorous testing, unlike digital health technology developed by clinical researchers, and may instead be designed to maximize user engagement. This has been termed the “commercialization gap” and it can lead to situations where DTC MHapps end up being popular despite being less effective [83,91]. The primary goal of such an app may be regular engagement, instead of efficacious treatment.

### 4.4. Access to and Adoption of Mobile Mental Health Interventions

Currently, there exist no consequences for marketing mobile health interventions containing inaccurate or non-evidence-based information, although calls to improve health app oversight and raise the standard of app development and clinical validation mechanisms are increasing [4]. The described issues give rise to opportunities for collaborations between industry and clinical researchers, with the goal of developing MHapps that are safe and effective, while also sufficiently engaging to ensure compliance and sustain therapeutic effect [69,92,93]. Such collaborations could infuse private app developments with the viewpoints and priorities of healthcare professionals, or vice versa, making interventions originating from research contexts more commercially viable and attractive.

Besides collaboration, a different approach to MHapp quality assurance may be to rethink the routes of access and accreditation of such interventions, to facilitate eventual integration with clinical practice. Curated, though limited, app libraries, such as Psyberguide [94] or the NHS App Library [95] aim to provide a solution to the unstructured and at times overwhelming access to (mental) health apps. Official regulatory bodies, such as the FDA and the European CE marking directives [96] list just 9 MHapps to date. App assessment tools, such as the APA framework [97] or the Mobile Application Rating Scale (MARS)/User MARS (uMARS) [98] put the onus on app users or their healthcare providers to assesses app quality.

Another recent example of MHapp access facilitation is the AppSalut Site, created by the Catalan Fundació TIC Salut Social [99]. This project aims to showcase apps in the field of health and social services, promoting health within the public. The catalogue allows prescription of certified apps by primary care doctors, and generated data can then be consulted by the professionals. A 5-month pilot study of this system validated the functionality of the platform and its compliance with data security regulations. However, it did not assess any form of clinical effectiveness [100].

In Germany, the Federal Parliament passed the Digital Healthcare Act (DiGA) in 2019, allowing digital health applications to be prescribed by either physicians or psychotherapists and reimbursed by statutory health insurance [101].

These types of projects and legislations allow us to start thinking about MHapps as something to be prescribed within a context of clinical guidance, similarly to common pharmaceutical interventions. This conceptual shift may also facilitate the construction of infrastructure, or a “pipeline” that allows for more robust clinical testing, validation, and evidence generation with the aim of being included within the prescribable pool of mobile applications. This could furthermore be coupled with continuous evidence-reporting and retrospective outcome assessment in individual patients, as well as across user populations, in cases where large-scale RCTs may not be feasible. These kinds of arrangements could be thought of as a type of market access agreement, which are subject to continued evidence development. Based on these outcome measurements, MHapps could be continually improved, and ineffective interventions could be weeded out. These types of structures should ideally work in concert with the development of standards, such as standardized health outcomes that should be consistently measured in studies assessing the effectiveness and safety of health apps, with specific adaptations for different therapeutic areas. Of course, such outcome measurement at individual patient level and subsequent incorporation of this data into the electronic health record, for example, would be ideal.

However, at a logistic and technological level, this might be a lofty goal to aspire to and lacking robust validation and accreditation is only one of the many stumbling blocks in the road toward incorporating MHapps in a broader healthcare context. Despite the precedent set by the DiGA, on a European level there exists no specific regulations on the use of digital therapeutics (DTx) [102]. Similarly, a dedicated FDA regulatory framework for software-as-a-medical-device (SaMD) solutions remains up in the air [103]. The European Data Protection Supervisor identifies various risks to data security in relation to DTx, such as constant observation of the patient or risk of data breaches, which make the development of appropriate legislation difficult [102]. Ensuring the security of large-scale healthcare data infrastructure, which offers appropriate levels of oversight, is a hugely complex task.

Furthermore, even considering that DTx solutions may undergo rigorous RCT validation, unlike traditional pharmaceuticals, they have the potential to be frequently updated after regulatory approval, a matter further complicated with the incorporation of AI technologies [103,104]. This means that regulatory pathways need modernization to account for the adaptive nature of DTx.

Lastly, a considerable issue in the adaptation of DTx is the lack of standardized payment and reimbursement frameworks [103]. Options may include licensing or value-based agreements, but without clear guidance on DTx financing within the various health insurance structures across Europe, prescribers and payers may be unable to transition away from traditional models and patients cannot access these therapeutic options.

While proper clinical validation and accreditation may certainly not be the only hurdle facing MHapps and DTx, it is an important step toward building an environment conducive to implement necessary frameworks, so that DTx interventions can be used safely and effectively.

### 4.5. Implications for Further Research and Policy

The message of this study is that clear, more robust evidence is necessary for the development and subsequent clinical implementation of MHapps. More outcomes-focused research is the crucial building block to harness the potential of mHealth in the treatment of mood disorders and mental health disorders in general.

Well-designed studies and the implementation of standardized outcome-monitoring could address concerns regarding effectiveness and safety and help overcome skepticism towards the systematic implementation of MHapps, both from the sides of healthcare professionals and users.

A concrete action toward holding MHapp and DTx products accountable to the same levels of scientific rigor that is expected of traditional pharmaceuticals could be the development and wide-spread use of app-assessment and accreditation tools. Having standardized assessment metrics to judge MHapp effectiveness and safety, which demand a certain type and quality of evidence, would not only ensure a product’s merit, but also help developers at the time of creating their apps. As with traditional medical products, knowing the requirements for approval helps guide the R&D process to gather all necessary data and substantiate a drug or device’s claims.

Entrenching such assessment tools in a formal authorization process undertaken by a national or international governing body cements the need for robust evidence if we want to start thinking about apps as prescribed therapeutic options or adjuvants. It is also clear that for this, DTx-specific legislation, approval pathways, and monitoring systems are necessary, which consider all the ways apps and digital solutions are distinct from traditional medicines and devices. The concept of app “administration” may lend itself more easily to appropriate regulatory oversight in terms of privacy and accountability, seeing as healthcare professionals are involved in the process. The authors believe that this is best encouraged and catalyzed through research and industry collaborations, which can capture the various relevant perspectives and needs. Involving diverse stakeholders, such as users, researchers, healthcare providers, and software developers in the creation of applications, as well as standards and best practices may best tackle the various issues effective MHapp-implementation still faces to date. Industry-based developers might find such corporations attractive if they can facilitate mHealth interventions reaching wider target populations and garner trust and a positive reputation with clinical professionals.

### 4.6. Limitations

There exists no gold standard for the systematic search and evaluation of mHealth interventions. Despite searching for apps designed to treat mood disorders such as depression, anxiety, dysthymia, and bipolar, resulting studies focused on various disorders that do not all fall into one of these categories, yet could be understood to be interrelated. Moreover, relying on the information that is publicly available through the Apple App and Google Play Stores carries some limitations, such as incomplete information, e.g., the lack of download information in the Apple App Store, or language specifications, and unstructured information organization. For the ease of our study, we developed a search-tool to extract the relevant data in a structured format. Our search was conducted in December of 2020, and considering that the app landscape is rapidly changing, conducting the same review at a later time might yield different results. Considering the above, we recognize that our results may not be reproducible, despite the transparency of our methods.

Furthermore, our review focused on a Spanish context. Despite searching app stores in the EU5 and USA, we did implement exclusion criteria that would filter out apps that were explicitly not available in Spanish. This was done because this study forms the basis of a larger research project, which aims to develop an app evaluation tool for use within the Spanish healthcare context. In addition, this study was not designed specifically in accordance with PRISMA guidelines. However, we did attempt to construct a robust rationale for the various inclusion and exclusion criteria we applied.

Lastly, we did not undertake a thorough examination of the functionalities of all apps, beyond the basic technical details, since this research limited itself more to understanding the evidence base instead of the effectiveness or adequacy of the individual apps in treating mood disorders. However, in further research it may be interesting to use the MARS/uMARS, for example, to evaluate other characteristics such as engagement, functionality, or information quality.

## 5. Conclusions

The use of digital technology in the treatment of mental health is an area of immense potential, especially considering the double-edged consequences of COVID-19; greater mental health burden accompanied by the increased facilitation of tele- and mHealth. Mental healthcare could be made more accessible and affordable, and stigma could be reduced through the effective use of MHapps. However, the lack of robust scientific evidence is continuously underscored, not only in the present study, but in many examinations of the current app-landscape. Finding ways to facilitate robust evidence-generation in a timely and cost-effective manner will remain a significant challenge. Moreover, it will always be necessary to ensure that compliance with meticulous empirical research standards is prioritized, over the potential appeals of producing a “hit” app. Here, research and industry collaborations, or innovative methodological approaches may offer some solutions, by incorporating diverse viewpoints to tackle issues such as producing efficacious apps, setting standards and best practices, and defining universally applicable empirical outcome measures. Additionally, appropriate regulatory oversight, especially when dealing with privacy and the protection of patient data, will be crucial.

## Figures and Tables

**Figure 1 ijerph-19-02186-f001:**
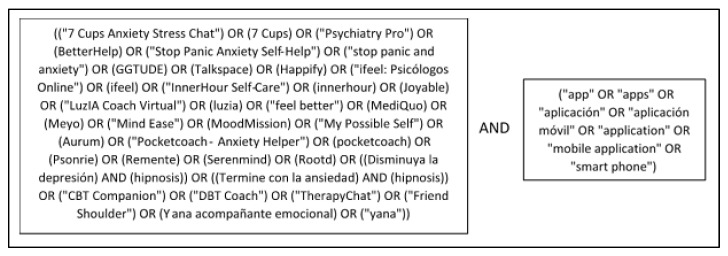
Search terms used for publication search of resultant apps.

**Figure 2 ijerph-19-02186-f002:**
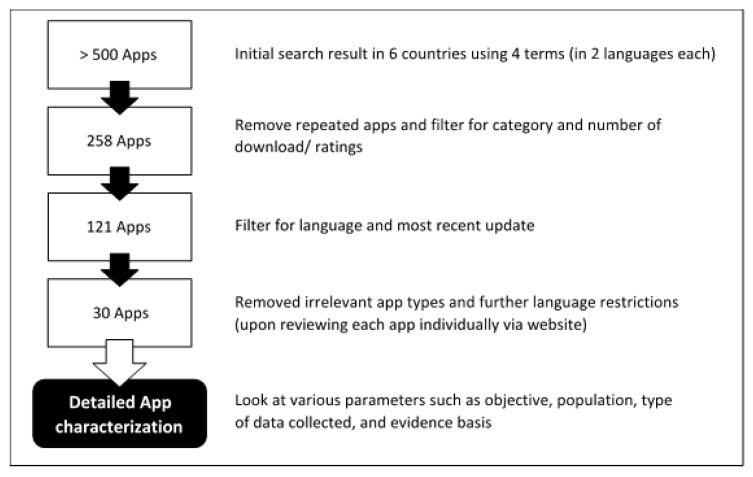
Flowchart of apps within the search.

**Table 1 ijerph-19-02186-t001:** Overview of app exclusion criteria.

List of Exclusion Criteria
Repeated app title Less than 10,000 downloads Not classified as one of the following genres ○ Health and Fitness ○ Lifestyle ○ Medical ○ Education Explicitly unavailable in Spanish Most recent update >1 year ago Irrelevant app type ○ Meditation/mindfulness only ○ Journaling/mood tracker ○ Diagnostic tests ○ Information only, no intervention

**Table 2 ijerph-19-02186-t002:** Resulting apps possessing available published evidence.

Applications with Published Evidence				
Name	Type	Publications	Objective	Population
7 Cups: Anxiety & Stress Chat	Chat and e-counselling	4 published articles [34,35,36,37]	Provide free active listening and paid online therapy	Various patient types, incl. stress, anxiety, depression, etc.
BetterHelp: Online Counseling & Therapy	E-counselling	1 published article [38]	Provide professional counselling, chat, and video messaging with therapist	Various patient types, incl. stress, anxiety, depression, etc.
DBT Coach	Cognitive behavioural therapy (CBT)	2 published articles [39,40]	Dairy, CBT/DBT (dialectical behavior therapy) exercises, peer groups	Various patient types, incl. stress, anxiety, depression, etc.
GGtude OCD Anxiety & Depression	CBT	6 published articles [41,42,43,44,45,46]	Applying CBT methods to break unhelpful thought patterns	OCD, anxiety, depression
Happify	Tracker, meditation, exercises	3 published articles [47,48,49]	Changing negative thought patterns, tracker, and activities/videos	Various patient types, incl. stress, anxiety, depression, etc.
Joyable: An AbleTo Program	CBT	1 published article [50]	2-month plan to deliver personalized CBT programs and support from coaches	Various patient types, incl. stress, anxiety, depression, etc.
MoodMission—Cope with Stress, Moods & Anxiety	Tracker, meditation, exercises	5 published articles [51,52,53,54,55]	Mood boosting activities, meditation, relaxation, exercise, affirmations, yoga, gratitude	Low mood, depression, stress, anxiety
Talkspace Therapy & Counseling	E-counselling	5 published articles [56,57,58,59,60]	Provide professional counselling, chat, and video messaging with therapist	Various patient types, incl. stress, anxiety, depression, etc.

**Table 3 ijerph-19-02186-t003:** Disease or treatment areas addressed in each publication.

Depression	8
Postpartum depression	(2)
Schizophrenia-spectrum disorders	1
Anxiety	7
Social anxiety	(1)
Borderline personality disorder	2
OCD *	3
Relationship OCD	(1)
Body image disorders	2
Suicide alert system	1
Self esteem	1
Loneliness	1
Response to lab-induced stressor	1
Mental wellbeing of patients with chronic diseases	1
Posttraumatic stress disorder	1

* OCD: obsessive-compulsive disorder.

## Data Availability

Not applicable.

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
