# Peer review of "Mobile Applications in Mood Disorders and Mental Health: Systematic Search in Apple App Store and Google Play Store and Review of the Literature"

_ijerph, 2022, doi:10.3390/ijerph19042186_

Round 1

Reviewer 1 Report

This manuscript explores mobile apps to treat mood disorders and mental health. I think it is a very interesting topic and of great interest to the scientific community and the general population.

I would like to make the following comments for a better understanding of the readers:

In the introduction I think it would be important to define depression, bipolar disorder, and dysthymia. I would also strongly recommend delving into Mobile health interventions, mentioning some studies that evaluate their impact on psychological variables in users or patients with mood disorders.

At the methodological level, exposition of results and discussion seems appropriate to me.

Author Response

Dear reviewer, 

thank you for your comments and positive feedback. 

Regarding your points for improvement for the introduction we have added the following: 

  • the introduction now contains definitions and diagnostic criteria of all the mentioned mood disorder
  • the introduction now contains three new references to systematic reviews/meta-analyses, which evaluate the positive impact of mental health apps on depressive and manic symptoms

Reviewer 2 Report

This study aims to understand and characterize the current landscape of mobile applications available in the Apple Store and Google Play Store for the mood disorders such as depression, bipolar disorder, and dysthymia. To do the study, the authors have developed a search tool based on keywords for the extraction of basic app information. In parallel the authors have searched PubMed for paper published on research done with the apps.

The theoretical background of this study contains the necessary information about the topic.

The results of the study showed that very few apps have study published about them and even fewer have some measures of effectiveness.

The very comprehensive material associated to this paper shows also that only of minority of the apps appear to be developed by mental health specialist (psychologist or psychiatrist).

The results of this study are also important for the public as they shown that mental health app available on the market generally don’t have scientific evidence behind them.

Page 2, line 81: Could you please provide a definition for "apptimism" it is intended as ‘’optimism related to app’’ or ‘’confident and hopeful that an APP will do as it is supposed to?’’ The part about ‘’Implications for future research and policy’’ should be developed to include some propose solutions based on the findings of the review.

Author Response

Dear reviewer, 

thank you for your comments and your positive feedback. 

In answer to your points for improvement, we have implemented the following: 

  • The introduction now contains an elaboration of what is meant by "apptimism", which refers to the optimistic outlook on the potential benefits of health app utilization
  • Furthermore, the section "implications for further research" has been expanded to include some discussion of more concrete suggestions/solutions. This includes a discussion of app assessment tools and the establishment of DTx-specific legislation and approval pathways, to entrench the need for evidence-based MHapps

Reviewer 3 Report

The Authors report on the state of the art of mobile health applications for mood disorders available in the market, with a neat focus on the scientific evidence supporting their clinical validation. As Authors claim – and I commend them for it - this endeavor lies in between the substantial lack of standard tools/procedures to empirically validate the clinical efficacy of these applications, and their great potential in backing treatment, especially in the case of mental health disorders.

The methodology for searching apps to the purpose has a sound rationale, and is described in detail.

The discussion highlights that, while a huge number of MH applications are developed and marketed, “the channels through which this is commonly done” (lines 236 onwards) do not intersect those clinical provision contexts in which such apps could be validated for effectiveness and safety. This conclusion is clear-cut and well contributes to the shaping of the state of art.

Overall, the paper conceptualization and implementation through subsequent sections is well designed, the research question is relevant both to the scientific community and to the governance issues connected with the marketing of mobile health apps for the treatment of mental disorders.

Nonetheless, I have two major remarks the Authors' consideration, which I hope will help to further strengthen the paper and its contribution.

The first one regards the explicit choice not to include apps that were not available in the Spanish language. Apparently the paper does not focus on the Spanish market, nor does it aim to draw country-specific conclusions. To me, this decision, though stated in the paper limitations, remains opaque and without a scientific rationale.

Moreover, although a consistent effort is spent in describing methodology, I was surprised at not finding a discussion of the principal findings in the context of Digital Therapeutics (Dtx). In the “Access to and adoption of mobile mental health interventions” section, for istance, Authors acknowledge that “calls to improve health app oversight and increase the standard of app development and clinical validation mechanisms” (lines 313-315). They also give examples  of “best practice” pipelines already implemented to this scope. Moreover, policy and legislation issues are hinted at also in the “Further research and policy” section”.

But issues such as, for instance,  the prescription of apps to be included in primary care go well beyond a development and evidence-based validation of mobile apps for the treatment of mental health disorders: they include organizational costs for health provision centers, besides the definition of a strategy for reimbursement options (in the private and/or public health systems), not to mention certification issues. At the same time, although the discussion still lingers between their scientific rationale, and public health legislation, the DTx broader discussion in the literature, points exacly at how empirical validation can support the adoption of such apps.

I believe that the discussion could be substantially enriched including these themes. And, possibily, also dealing with the difficulties in characterizing this landascape due to country-specific legislations - which is, at least in the EU context, is a major issue.

Author Response

Dear reviewer, 

thank you for your positive and constructive feedback. 

In answer to your points for improvement, we have implemented the following:

  • the limitations section now includes a more detailed explanation for filtering for Spanish language apps. This is because this study forms the basis of a larger research project, which aims to develop an app evaluation tool for use within the Spanish healthcare context.
  • Furthermore, the section "access to and adoption of mobile mental health interventions" now contains a more detailed discussion of MHapps within the context of digital therapeutics and the hurdles in its adoption that go beyond validation and accreditation, which include lack of regulatory frameworks, security issues, and inappropriate reimbursement models. 

Reviewer 4 Report

This study was done to explore mobile apps available for mood disorders. The authors searched scientific publications for commercial apps.  The introduction explains that this is important since everyone uses mobile phones so much and apps are downloaded often.  It's important to see what evidence is used to create and modify a commercial app.  The design/methods is written well.  I appreciate showing the inclusion/exclusion of apps.  I think the authors could probably be more clear about how something was excluded.  Perhaps make a table with reasons why app was excluded.  

The authors wrote a good discussion, although I do feel it is very wordy, and would be helpful to have some tables to explain the text better. One table that I think would be helpful is to have explain what evidence there was for certain apps - for example, was it a pilot study, RCT, how big of a study, and how robust the evidence? And what did the study results show? Do the apps get marketed even if the study found the app to not be helpful?  Perhaps, a comment about what results showed and whether app still was marketed would be helpful. 

Typo in Applications with published guidance: for GGtude OCD Ansceity and Depression - CBT to "beak" should be "break"

Author Response

Dear reviewer, 

thank you for your feedback and your positive comments. 

In answer to your points for improvement, we have implemented the following: 

  • The typo in table 1 has been corrected
  • We've added a brief table outlining the exclusion criteria in the methods section 
  • regarding the study details for each mentioned publication, this information was included already in supplementary table number 4. We have added more clear references to this table within the main text (discussion section "principal findings") and briefly mention the described the findings of the clinical studies. 

Round 2

Reviewer 3 Report

The Authors fully addressed the suggestions. Thank you.